# Polysaccharide of *Atractylodes macrocephala* Koidz Alleviates Cyclophosphamide-Induced Thymus Ferroptosis in Gosling

**DOI:** 10.3390/ani12233394

**Published:** 2022-12-02

**Authors:** Xiangying Zhou, Nan Cao, Danning Xu, Yunbo Tian, Xu Shen, Danli Jiang, Yunmao Huang, Wanyan Li, Bingxin Li

**Affiliations:** 1College of Animal Science & Technology, Zhongkai University of Agriculture and Engineering, Guangzhou 510225, China; 2Guangdong Province Key Laboratory of Waterfowl Healthy Breeding, Guangzhou 510225, China

**Keywords:** polysaccharide of *Atractylodes macrocephala* Koidz, cyclophosphamide, ferroptosis, immunosuppression, thymus, gosling

## Abstract

**Simple Summary:**

During the brooding stage, the goslings are susceptible to various stresses, damaging the thymus and decreasing immune function. Ferroptosis is cell death caused by iron-ion-dependent lipid peroxidation, which has been reported to be closely related to organismal immunity. The polysaccharide of *Atractylodes macrocephala* Koidz (PAMK) has antioxidation and immunomodulatory effects. Therefore, we used CTX to construct an immunosuppression model for goslings to explore the mechanism by which PAMK alleviates cyclophosphamide (CTX)-induced ferroptosis in thymocytes and to provide a basis for a more profound elucidation of the immunomodulatory mechanism of PAMK. It was found that PAMK had a significant alleviating effect on CTX-induced thymus damage and thymocyte ferroptosis in goslings. Therefore, PAMK can be used as a natural alternative to antibiotics as a feed additive for immunomodulatory effects on goslings.

**Abstract:**

The present study aimed to explore the mechanism by which PAMK alleviates cyclophosphamide (CTX)-induced ferroptosis in thymocytes. One-day-old goslings were divided into four groups (10 goslings/group). The CON and CTX groups were fed a basic diet. The PAMK and CTX + PAMK groups were fed the basic diet mixed with PAMK (400 mg/kg). Moreover, the CTX and CTX + PAMK groups were given a daily injection of 40 mg/kg BW of CTX (at 19, 20, and 21 days of age). On the other hand, the CON and PAMK groups were given 0.5 mL of sterilized saline into the leg muscle (at 19, 20, and 21 days of age). The goslings were fed for 28 days. The ferroptosis pathway was enriched in transcriptome sequencing. Compared to the CON group, the thymus in the CTX group underwent injury, and the mitochondria of thymocytes showed features of ferroptosis. PAMK treatment alleviated ferroptosis in thymocytes and thymus injury, and CTX-induced elevated levels of oxidative stress and iron content restored GPX4 protein expression (*p* < 0.05) and inhibited the CTX-induced activation of the ferroptosis pathway (*p* < 0.05). Conclusively, PAMK could reduce thymus injury by alleviating CTX-induced thymocyte ferroptosis in gosling to alleviate the immunosuppression caused by CTX in the organism.

## 1. Introduction

Ferroptosis is cell death caused by oxidative damage from iron-ion-dependent lipid peroxidation, leading to iron metabolic dysfunction and massive lipid peroxidation [1]. Ferroptosis mainly involves the accumulation of reactive oxygen species (ROS) and intracellular iron, which causes an imbalance in the balance of intracellular lipid ROS production and degradation, ultimately causing oxidative stress. Many studies have shown that ferroptosis is strongly associated with the immunity of the body and that the activity of innate and adaptive immune cells is regulated by ferroptosis [2,3,4].

The thymus, as a central immune organ for T cell development and differentiation, has a significant regulatory role in the immune function of the organism [5]. During the brooding stage of goose, due to the imperfection of immune function and the influence of the aquatic breeding environment, goslings are susceptible to various stresses, resulting in damage to the thymus gland and a decrease in immune function, leading to increased morbidity and mortality in goslings. Therefore, immunosuppression caused by these stresses is a significant causal factor in the morbidity of goslings [6]. Although the arid breeding mode can avoid the hazards caused by the bad environment of aquatic breeding, at present, for the large-scale breeding of geese, the completely arid breeding method is still in the exploratory stage. To simulate the process of immunosuppression due to stress in goslings, we used CTX to construct a model of immunosuppressed goslings. CTX, a widely used drug in the preparation of animal immunosuppression models, also leads to oxidative stress, inflammation, and apoptosis [7]. Most studies have found that CTX induces apoptosis [8,9]. Only one report has suggested that CTX induces ferroptosis in tumor cells [10]. However, it is unclear whether CTX causes ferroptosis’s development in avian species and what role ferroptosis plays in CTX-induced immunosuppression.

PAMK is the main active component of the Chinese medicine *Atractylodes macrocephala*, which has been recorded in the Chinese medical classic Shengnong’s herba. Studies have shown that PAMK can promote the development of immune organs, regulate the proliferation and differentiation of immune cells, positively regulate the immune response capacity of the body, and alleviate the immunosuppression and immune organ damage caused by harmful stimuli, maintaining normal immune organ function and the homeostasis of the immune system [11,12]. Our preliminary study found that PAMK could activate T lymphocytes in the thymus of goslings, maintain the balance of cytokine secretion in vivo, alleviate immunosuppression caused by CTX [13,14], and also alleviate splenic immune dysfunction in chickens caused by heat stress by enhancing mitochondrial function, inhibiting apoptosis, and reducing oxidative stress [15].

In this study, we constructed a model of immunosuppressed goslings using CTX, added PAMK for treatment. To explore the mechanism by which PAMK alleviates CTX-induced thymus injury, we first conducted transcriptome sequencing of the thymus and enriched the ferroptosis pathway. Based on transcriptome sequencing, we speculated that PAMK could alleviate CTX-induced ferroptosis in the thymocytes of goslings. Therefore, we observed the thymus histology and ultramicroscopic morphology and detected ferroptosis marker GPX4 and thymus iron content, as well as oxidative stress levels and genes related to the ferroptosis pathway. The mechanism of PAMK alleviating CTX-induced ferroptosis in thymocytes was preliminarily explored to provide a basis for more in-depth elucidation of the immunomodulatory mechanism of PAMK and a theoretical basis for the further development of natural plant additives.

## 2. Materials and Methods

### 2.1. Animal Experiments

All goslings were purchased from Guangdong QingyuanJinyufeng Goose Co., Ltd., Qingyuan, China. Forty 1-day-old Magang goslings (half male and half female) were randomly divided into four groups of 10 goslings each and pre-fed for 3 days. The CON and CTX groups were fed a basic diet. The PAMK and CTX + PAMK groups were fed the basic diet mixed with PAMK (400 mg/kg). At 19, 20, and 21 days of age, the CTX and CTX + PAMK groups were given a daily injection of 40 mg/kg BW of CTX (Baster, Berlin, Germany) into the leg muscle. On the other hand, the CON and PAMK groups were given 0.5 mL of sterilized saline into the leg muscle (Figure 1). The doses of CTX and PAMK were determined based on previous studies [12]. All goslings were allowed to feed and drink freely. Thymuses were collected at day 28 of age. Ethical approval for this experiment was obtained from the Zhongkai University of Agricultural and Engineering under the approved protocol NO. 20191201.

### 2.2. Transcriptome Sequencing

Transcriptome sequencing was performed by BGI Genomics Co., Ltd., (Wuhan, Chian). Total RNA was extracted from the thymus using TRIzol reagent (15596026, Ambion, Austin, TX, USA). High-quality RNA was used to construct sequencing libraries and analyze the enrichment of differentially expressed genes (DEGs). First, data were filtered and assembled using HISAT2 software, and the expression of genes and transcripts were calculated using RSEM software. The interception criteria for DEGs were |logFC| ≥ 1 and adjusted *p*-value < 0.05. All DEGs were visualized as heat maps and volcano maps using R packages: pheatmap and ggplot2, respectively. Gene ontology (GO) enrichment analysis, including biological processes, cellular components, and molecular functions, were performed using Blast2GO software to determine the biological roles of the DEGs.

### 2.3. Real-Time Quantitative PCR Assay

Total RNA was extracted from the thymus of goslings using TRIzol reagent according to the manufacturer’s reagent instructions (15596026, Ambion, USA) and reverse-transcribed using reverse-transcription reagent (RR036A, Takara, Dalian, China). Subsequently, cDNA templates from the samples were amplified with SYBR Green (A25742, Applied Biosystems, Waltham, MA, USA). Primer sequences were designed according to the NCBI database (Table 1). We used the ABI PRISM 7500 detection system (Applied Biosystems, USA) to detect the relative mRNA expression of the genes. In this assay, β-actin was used as an internal reference gene, and the mRNA expression of each gene was calculated using the 2^−∆∆ct^ relative quantification method.

### 2.4. Western Blot Analysis

We collected thymus for protein extraction using RIPA lysis buffer (Beyotime, Shanghai, China) supplemented with protease inhibitors (Beyotime, Shanghai, China). The protein from the thymus was separated by electrophoresis. Transfer the protein to the PVDF membrane at 4 °C. After blocking with 5% skimmed milk at room temperature for 2 h, the membranes were incubated with GPX4 primary antibody (ab40993, Abcam, Waltham, MA, USA) and GAPDH primary antibody (60004-1-Ig, Proteintech, Wuhan, China) at 4 °C overnight. After incubation with peroxidase-conjugated secondary antibodies (PR30011 and PR30012, Proteintech, Wuhan, China) for 1 h at room temperature. The signal was detected on an ImageQuant LAS 500 (GE, Chicago, IL, USA) using the ECL Kit (Biosharp, Hefei, China). GAPDH expression served as a loading control for quantification.

### 2.5. Thymus Histology

We cut the paraffin-fixed blocks into consecutive coronal sections 5–6 μm thick. For routine histological examination, paraffin sections were stained with HE. The sections were scanned with a section scanner (NanoZoomer S360, Hamamatsu Photonics, Hamamatsu, Japan), and the HE-stained scanned sections were analyzed with CaseViewer. We observed changes in HE-stained sections of the thymus, selected three sections from each group with 10 randomly selected fields of view (200×), and measured the thickness of the thymic cortex area in each field of view for statistical analysis.

### 2.6. Ultramicroscopic Morphology Observation

Each thymus was divided into 1 mm three pieces and then fixed with 2.5% glutaraldehyde at 4 °C. Ultrathin sections of 50–70 nm thickness were prepared and stained with uranyl acetate (22400, EMS, Hatfield, PA, USA) and lead citrate (19314, TED PELLA, Redding, CA, USA). Samples were observed with a transmission electron microscope (JEM-1400, JEOL, Mitaka-shi, Japan) and with a scanning electron microscope (Hitachi S3000N, Tokyo, Japan). The ratio of dead cells to live cells was calculated at 3000×.

### 2.7. Immunofluorescence Staining

Thymus sections were dewaxed in xylene and hydrated with a gradient ethanol solution. Antigen repair was performed on the sections using EDTA antigen repair buffer (G5012, Servicebio, Wuhan, China). Endogenous peroxidase was blocked with a 3% hydrogen peroxide solution. The thymus sections were blocked in BSA (BS114, Bioshap, Hefei, China) for 1 h and and incubated with the GPX4 primary antibody (ab40993, Abcam, USA) at 4 °C overnight. The secondary antibody was used at a 1/1000 dilution for 1 h at room temperature. DAPI was used to stain the cell nuclei. Images were acquired by a fluorescent inverted microscope (Carl Zeiss, Jena, Germany). Anti-fluorescence quenching mounting tablets (Servicebio, Wuhan, G1401) were used for mounting. Image J was used to calculate the GPX4 fluorescence.

### 2.8. ROS Assay

The content of fresh thymus intracellular ROS was measured using the ROS detection kit (Beyotime, Shanghai, China). Fresh thymus from goslings was collected and ground on a 70 μm cell sieve to filter the cells and into PBS. Centrifugation was performed at 1600 rpm for 5 min, and the supernatant was discarded. Add 200 μL DCFH-DA staining solution, adjust the concentration of cells to 5 × 10^6^ cells/mL, mix well, and incubate for 15 min away from light. The cells were washed with PBS, centrifuged at 1600 rpm for 5 min, discarded the supernatant, and resuspended in 300 μL PBS. The resuspended cells were detected using a flowmeter (BD Biosciences, San Jose, CA, USA).

### 2.9. Glutatione (GSH), Malonedialdehyde (MDA), and Tissue Fe Level Assays

GSH (A006-2-1), MDA (A003-1-2), and tissue Fe (A039-2-1) level assays were performed using the kit (Nanjing Jiancheng Institute of Biological Engineering, Nanjing, China) according to the manufacturer’s instructions.

### 2.10. Statistical Analysis

The experimental data were analyzed by one-way ANOVA using GraphPad Prism 7.0. The differences between the four groups were compared by Tukey’s multiple comparison method, and *p* < 0.05 were considered as statistically significant.

## 3. Results

### 3.1. Transcriptome Analysis Reveals PAMK Might Regulate Ferroptosis Pathway

To clarify the mechanism of action of CTX on thymus injury and PAMK on thymus protection, we first performed RNA sequencing of CTX and the thymus of CTX and PAMK co-treated goslings to systematically investigate the biological process of the effect of PAMK and CTX. Figure 2A is a heatmap of DEGs clustering relationship between CTX and PAMK + CTX groups, and Figure 2B shows the volcano map with 335 upregulated genes and 440 downregulated genes. In the GO enrichment analysis, there is an enrichment of DEGS in cellular process, immune system process, and antioxidant activity (Figure 2C), which is what we focus on. Among them, ferroptosis involves not only cell process but also antioxidant activity. In Figure 2D, the enrichment of DEGs in the ferroptosis signaling pathway is depicted, mainly involving the cystine/glutamate antiporter system (system xc-), arachidonic acid (AA) metabolism pathway, and the iron transport and metabolism pathway. The PAMK + CTX group showed higher gene expression of SLC7A11, GSS, and Ferroportin (FPN1) and lower gene expression of ACSL4, TF, STAEP3, VDAC2, and VDAC3 than the CTX group, demonstrating that PAMK and CTX might involve regulating cellular ferroptosis in the thymus of goslings, while PAMK might inhibit ferroptosis.

### 3.2. PAMK Alleviates CTX-Induced Thymus Injury, Cell Death, and Mitochondrial Damage

The above results suggest that CTX leads to the activation of the ferroptosis pathway and that PAMK has an inhibitory effect on this pathway. Therefore, we observed the microstructure of the thymus to analyze whether the activation of the ferroptosis pathway caused thymus injury. HE staining results (Figure 3E,F) showed that thymocytes in the CON group were neatly arranged, with normal morphology, clear cortex and medullary demarcation, and thick cortex (Figure 3H); the morphology and arrangement of cells in the PAMK group were not significantly different from those in the CON group. The cells in the PAMK + CTX group were more closely arranged than those in the CTX group, and the cortex and medullary boundaries were clear. For the quantification of the mean thickness of the thymic cortex, it was significantly lower in the CTX group compared to the CON group and significantly increased in the PAMK + CTX group compared to the CTX group.

In scanning electronic microscopy (Figure 3D), CTX caused lymphocyte atrophy and increased connective tissue, while PAMK restored cell morphology and reduced connective tissue proliferation. In transmission electron microscopy (Figure 3A–C), the cell morphology and chromatin distribution were normal in the CON and PAMK groups. The CTX group showed chromatin borders, the disappearance of nuclei, irregular cell morphology with cavities and apoptotic bodies, and an increased number of dead cells (Figure 3G). It is noteworthy that the mitochondria in the CTX group became smaller and wrinkled, that their number decreased, and that the mitochondrial cristae disappeared, which are characteristic of ferroptosis. Cell morphology in the PAMK + CTX group was more regular; mitochondrial cristae increased, and the mitochondrial morphology approached normal. It proved that apoptosis and ferroptosis might be involved in CTX-induced thymus injury in goslings and that PAMK could alleviate this phenomenon.

### 3.3. PAMK Alleviated the Decline of GPX4 in the Thymus

To further clarify the involvement of ferroptosis in CTX-induced thymus injury, we detected the protein expression of GPX4 in thymus by immunofluorescence (Figure 4). The results showed that the protein level of GPX4 was significantly downregulated in the thymus after CTX injection. There were no significant differences in the protein levels of GPX4 in the thymus of the CON, PAMK, and PAMK + CTX groups. This indicates that CTX stimulation led to a significant decrease in the protein level of GPX4 in the thymus of the goslings, resulting in a weakening of the GPX4 catalytic peroxidation of lipid substrates, the accumulation of lipid ROS, and, eventually, ferroptosis, while PAMK could alleviate ferroptosis by increasing the protein expression of GPX4. Furthermore, immunofluorescence pictures showed that GPX4 expression was mainly concentrated in the thymus corpuscles and that the thymic corpuscle was also decreased in the presence of immunosuppression.

### 3.4. PAMK Alleviated CTX-Induced Elevated Levels of Oxidative Stress and Iron in the Thymus

The occurrence of ferroptosis is closely related to the level of antioxidants and iron ions. Therefore, after confirming the occurrence of ferroptosis, we measured ROS, MDA, GSH, and Fe content (Figure 5). Compared to the CON group, ROS levels and the content of MDA and iron were significantly increased (*p* < 0.05), and the content of GSH was significantly decreased (*p* < 0.05) in the CTX group. In the PAMK + CTX group, the content of GSH was significantly increased (*p* < 0.05), and ROS levels were significantly decreased (*p* < 0.05), compared to the CTX group. As for the content of MDA and iron, although there was no significant difference in the PAMK + CTX group compared to the CTX group (*p* > 0.05), the trend was opposite to that of the CTX group. This suggests that ferroptosis may be involved in CTX-induced thymus injury in goslings by increasing levels of oxidative stress and iron in the thymus, which can be alleviated by PAMK.

### 3.5. PAMK Blocked the CTX-Induced Activation of Ferroptosis Pathway in the Thymus

Finally, we analyzed the specific changes in the ferroptosis pathway. We examined the relative mRNA expression of ferroptosis pathway genes and the protein expression of GPX4, a marker protein of ferroptosis (Figure 6). Among the genes involved in promoting ferroptosis, the mRNA expression of ACSL4, COX-2, TFR1, VCAD2/3, TF, STEAP3, and NRF2 was significantly higher in the CTX group compared with the CON group (*p* < 0.05), except for Hmox-1. The mRNA expression of the above genes was significantly lower in the PAMK + CTX group than in the CTX group (*p* < 0.05), except for the NRF2.CON group (*p* < 0.05). CTX stimulation did not cause a significant decrease in the mRNA of GPX4 but could cause a significant decrease in the protein expression of GPX4 (*p* < 0.05). Although the change in FPN1 mRNA expression between the CTX and PAMK + CTX groups was insignificant, the expression in the PAMK + CTX group returned to a level similar to that of the CON group. The above results suggest that PAMK can alleviate CTX-induced ferroptosis in thymocytes by blocking the activation of the ferroptosis pathway.

## 4. Discussion

Ferroptosis is a new form of programmed cell death with iron-dependent properties, whose main features include lipid reactive oxygen species accumulation, iron ion accumulation, and lipid peroxidation. The metabolites of CTX after hepatic biotransformation exert a suppressive effect on the immune function of the body [16], bringing a series of impairments in immune function and oxidative damage to the body [17,18]. It has been shown that CTX can induce ferroptosis through the NRF2/HMOX-1 pathway [10]. Drugs that also have immunosuppressive effects, such as dexamethasone, partially involve ferroptosis in the T-cell ablation induced in zebrafish [19]. Based on the above studies, we speculate that ferroptosis may be involved in CTX-induced thymus injury in goslings, leading to immunosuppression in the gosling organism. Many studies have found that herbal medicines can alleviate cellular damage and metabolic disorders caused by other diseases by inhibiting ferroptosis [20,21,22]. As an immunomodulator, PAMK can improve the immunity of the organism, reduce stress damage, and effectively protect the goslings from epidemics [23,24]. Therefore, we performed the transcriptome sequencing of the thymus of goslings in the CTX and PAMK + CTX groups, and among the differentially expressed genes, we enriched the ferroptosis pathway. Meanwhile, PAMK alleviated CTX-induced ferroptosis in thymocytes through the regulation of genes related to system xc-, the AA metabolism pathway, and the iron transport and metabolism pathways.

Studies on the cytotoxicity of CTX have mainly focused on apoptosis [7,25]. However, many studies have found that ferroptosis and apoptosis are closely linked, that apoptosis can be converted into ferroptosis under certain conditions, and that ferroptosis promotes cell sensitivity to apoptosis [26,27]. Hu et al. found that apoptosis, pyroptosis, and ferroptosis together induced an immunosuppressive hepatocellular carcinoma microenvironment and γδ T-cell imbalance [28]. Our HE, TEM, and SEM results showed that CTX causes thymus injury, which is mainly manifested by the indistinct cortex and medullary demarcation of the thymus and reduced cortical thickness. Thymocytes were disordered in arrangement and underwent apoptosis. These findings are consistent with previous studies [29,30]. Ferroptosis is morphologically manifested mainly by mitochondrial atrophy, the reduction or disappearance of mitochondrial cristae, increased mitochondrial membrane density, and the rupture of the outer mitochondrial membrane [31]. Our electron microscopic results clearly showed that the mitochondria of cells in the CTX group exhibited ferroptosis. All of these conditions were alleviated after the addition of PAMK. The above indicates the involvement of not only apoptosis but also ferroptosis in thymus injury caused by CTX. PAMK may alleviate CTX-induced thymus injury by reducing apoptosis and ferroptosis in thymocytes, and this study focused on ferroptosis.

When ROS accumulates in excess in cells, membrane polyunsaturated fatty acids are easily oxidized by ROS and produce MDA, for example, causing ferroptosis [32]. GPX4 usually catalyzes the removal of lipid peroxides such as ROS, and this process requires glutathione (GSH) as a cofactor [33]. Matsushita et al. found that GPX4 deficiency-induced T cell death was caused by a lipid peroxidation-mediated pathway that involves ferroptosis [34]. Our results showed that CTX significantly decreased the protein levels of GPX4, suggesting that CTX is involved in stimulating ferroptosis in the thymus. In contrast, the co-treatment of PAMK and CTX restored the protein expression of GPX4 to a normal level, suggesting that PAMK can effectively alleviate the thymus ferroptosis involved in CTX. Surprisingly, we found that GPX4 is predominantly expressed in the thymic corpuscle. In the presence of immunosuppression, the number of thymic corpuscles decreases, and the thymus lacking the thymic corpuscle cannot cultivate T cells [35]. Therefore, we speculate that GPX4 may have an essential role in T cell development and differentiation. In our study, we found that CTX treatment resulted in the accumulation of ROS, increased MDA content, the depletion of GSH, and the inactivation of GPX4, which led to a decrease in the antioxidant capacity of the organism and resulted in the development of ferroptosis. PAMK could alleviate ferroptosis in thymocytes by increasing the protein level and antioxidant capacity of thymus GPX4 in gosling.

Iron homeostasis has been shown to play a critical role in the innate immunity of the organism [36,37]. Host cells can use iron to produce ROS to clear microbes and promote cell survival [38]. However, the excessive accumulation of free iron in cells can promote the excessive accumulation of ROS through the Fenton reaction, leading to the ferroptosis of cells [39]. Meanwhile, intracellular iron accumulation will, in turn, promote intracellular microbial infection [40]. Our study found that CTX stimulation caused a significant increase in iron content in the thymus of gosling, leading to an imbalance in iron homeostasis; co-treatment of PAMK with CTX resulted in a decrease in iron content in the thymus. Our study found that CTX stimulation caused a significant increase in iron content in the thymus of goslings, leading to an imbalance in iron homeostasis; the co-treatment of PAMK with CTX resulted in a decrease in iron content in the thymus. This demonstrates that PAMK can alleviate CTX-induced thymocyte ferroptosis in the thymus by reducing the iron content of the thymus, thereby reducing ROS production, and can restore the immune function of the organism by regulating iron homeostasis.

GPX4 plays a very central role in inhibiting ferroptosis, but we found that CTX does not inhibit GPX4 transcription but rather GPX4 protein expression, which is consistent with previous studies [41]. The new study found that GPX4 can form non-covalent bonds with CTX, demonstrating that CTX is a potential ferroptosis inducer [42]. Therefore, it is inevitable that CTX can inactivate the GPX4 protein by target binding to it, leading to a reduction in the antioxidant capacity of the body, allowing an excessive accumulation of ROS and leading to ferroptosis. Based on our previous study [24] and transcriptome sequencing enriched for differentially expressed genes in the ferroptosis pathway, we also examined the mRNA expression of SLC7A11, VDAC2/3, AA metabolism, NRF2/HMOX-1 axis, and genes related to iron transport metabolism. Wang et al. found that the genetic deletion or mutation of SLC7A11 inhibited GSH synthesis, leading to increased tissue lipid peroxidation and ferroptosis [43], which is similar to our findings. The mRNA expression of SLC7A11 in the thymus is decreased by CTX, which affects GSH synthesis and contributes to GPX4 inactivation. As a critical enzyme in AA production, cyclooxygenase 2 (COX-2) is involved in mediating ferroptosis [44]. COX-2 expression is, in turn, regulated by Acyl-CoA synthase long-chain family member 4 (ACSL4) [45]. In the present study, PAMK was found to inhibit the CTX-induced overexpression of the ACSL4/COX-2 axis. Previous studies demonstrated that NRF2-derived HMOX-1 could neutralize accumulated ROS when HMOX-1 expression is moderately activated; however, the overactivation of HMOX-1 increases iron pools, leading to ROS overload and subsequent oxidative cell death [46,47]. We found that PAMK alleviated the CTX-induced overexpression of HMOX-1 but did not have an inhibitory effect on the overexpression of NRF2. Excess intracellular iron underlies ferroptosis, and iron binds to transferrin (TF) in the form of Fe3+ and then enters the cell via transferrin receptor 1 (TFR1) [48,49]. FPN1 is the only known ferric ion efflux protein and is involved in the regulation of ferroptosis. Zhang et al. found that FPN1 affects macrophage iron release and plays a vital role in regulating the innate immune response [50]. The present study shows that PAMK can reduce excess iron in the thymus by regulating genes involved in iron transport and metabolism, thereby alleviating ferroptosis. It has been shown that the activation of voltage-dependent anion channel 2/3 (VDAC2/3) blocks mitochondrial depolarization, promotes ROS release, and induces ferroptosis [51,52]. We found that PAMK could inhibit the CTX-induced overexpression of VDAC2/3, thereby suppressing ROS release. The above studies suggest that PAMK affects cellular ferroptosis in the gosling thymus by regulating the expression of several ferroptosis-related genes.

## 5. Conclusions

In conclusion, PAMK could reduce thymus injury by alleviating CTX-induced thymocyte ferroptosis in gosling to alleviate the immunosuppression caused by CTX in the organism. Meanwhile, PAMK could co-mitigate ferroptosis in thymocytes by regulating the expression of critical genes of ferroptosis, restoring GPX4 protein expression, improving antioxidant capacity, and reducing iron overaccumulation (Figure 7).

## Figures and Tables

**Figure 1 animals-12-03394-f001:**
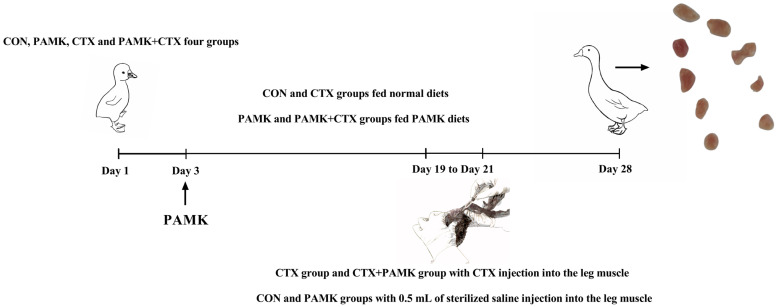
Schematic outlines of the experimental approaches tested in goslings.

**Figure 2 animals-12-03394-f002:**
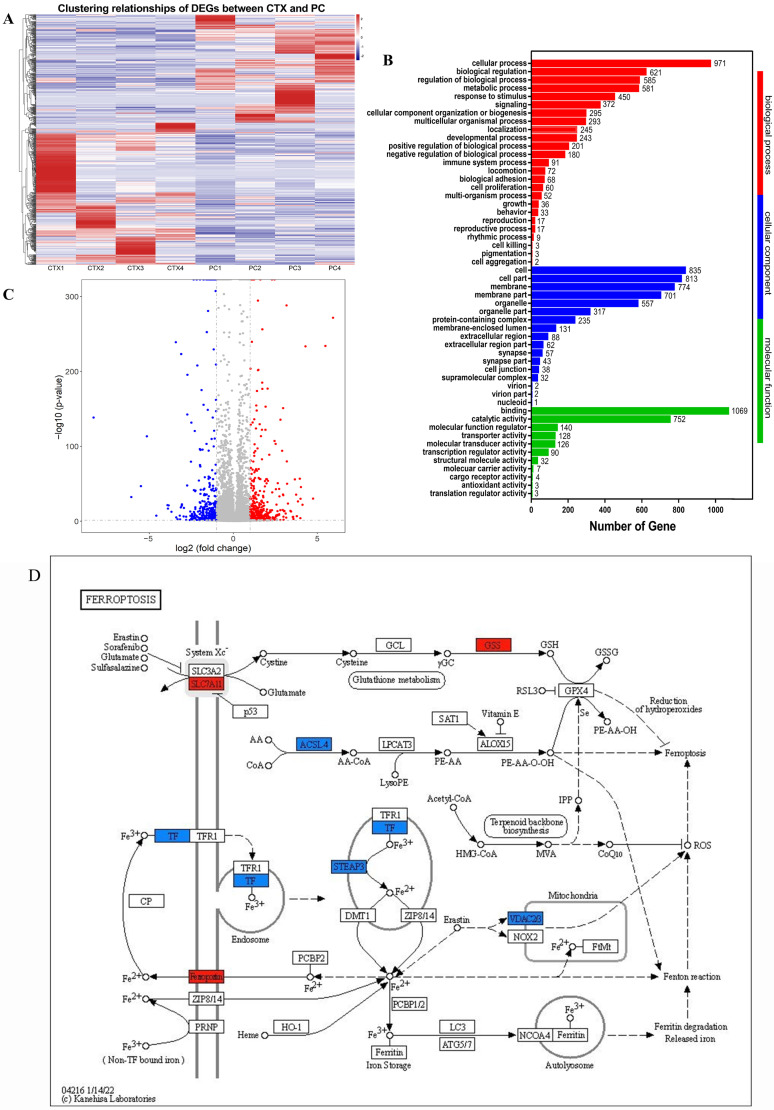
The ferroptosis pathway was involved in the thymus of CTX-treated goslings. (**A**) A heatmap of DEGs clustering relationship between CTX and PAMK + CTX groups; CTX indicates the CTX group; PC indicates PAMK + CTX group; (**B**) GO enrichment analysis of DEGs; (**C**) volcano plots visualizing DEGs with a cut-off criterion of excluding genes with 0 expression (red dots represent upregulated genes; blue dots represent downregulated genes); (**D**) Ferroptosis signaling pathways and associated genes (red indicates upregulated genes; blue indicates downregulated genes). *n* = 4.

**Figure 3 animals-12-03394-f003:**
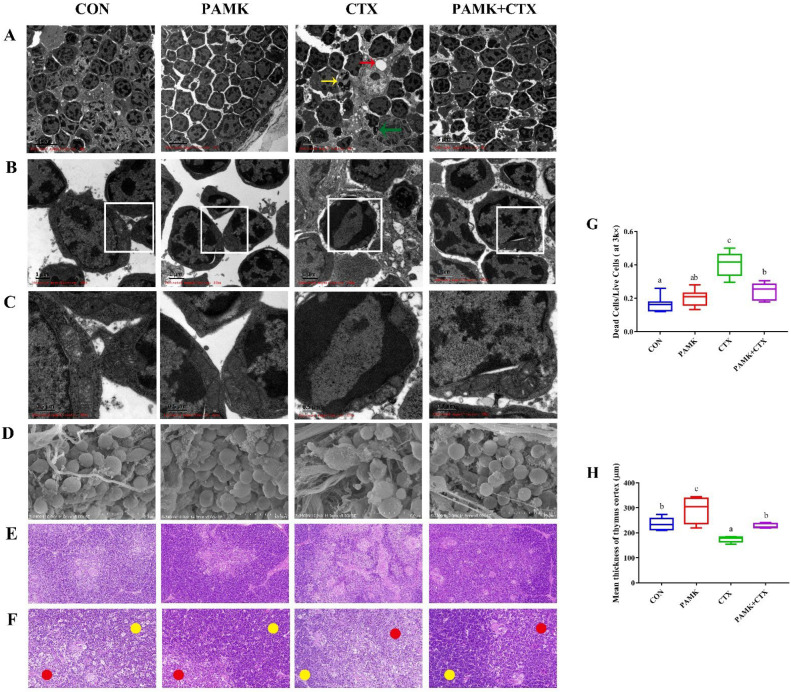
Effects of PAMK on histology and the ultramicroscopic morphology of thymus treated with CTX. (**A**) Transmission electron microscopy (TEM; 3000×) of the thymus. The red arrows indicate cell cavity; the yellow arrows indicate apoptotic cells; the green arrows indicate the apoptosis body. (**B**) Transmission electron microscopy (TEM; 10,000×) of the thymus. (**C**) Transmission electron microscopy (TEM; 30,000×) of the thymus. (**D**) Scanning electron microscopy (SEM; 5000×) of the thymus. (**E**) HE staining of the thymus (200×). (**F**) HE staining of the thymus (600×); red circles indicate the thymic medulla; yellow circles indicate the thymic cortex. (**G**) Ratio of dead cells to live cells in a single field of view at SEM (3000×); (**H**) Quantification of the mean thickness of the thymic cortex. Data are expressed as min to max, *n* = 3. Different letters indicate *p* < 0.05, significantly different.

**Figure 4 animals-12-03394-f004:**
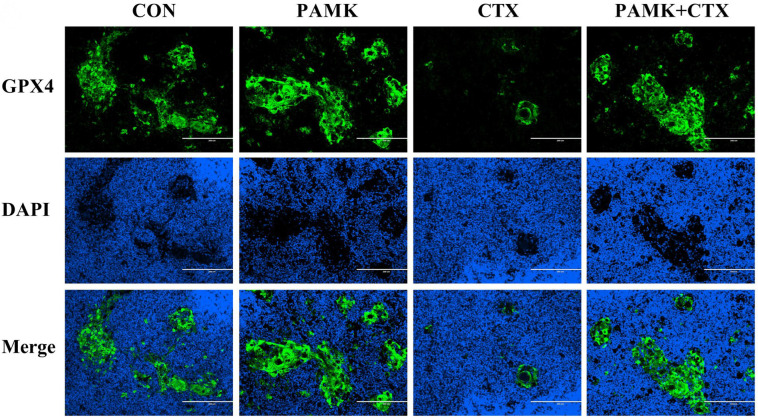
Effects of PAMK on the protein expression of GPX4 in thymus treated with CTX. Images of immunofluorescence staining for GPX4. Representative images from experiments with similar results are shown (scale bar = 200 μm). *n* = 3.

**Figure 5 animals-12-03394-f005:**
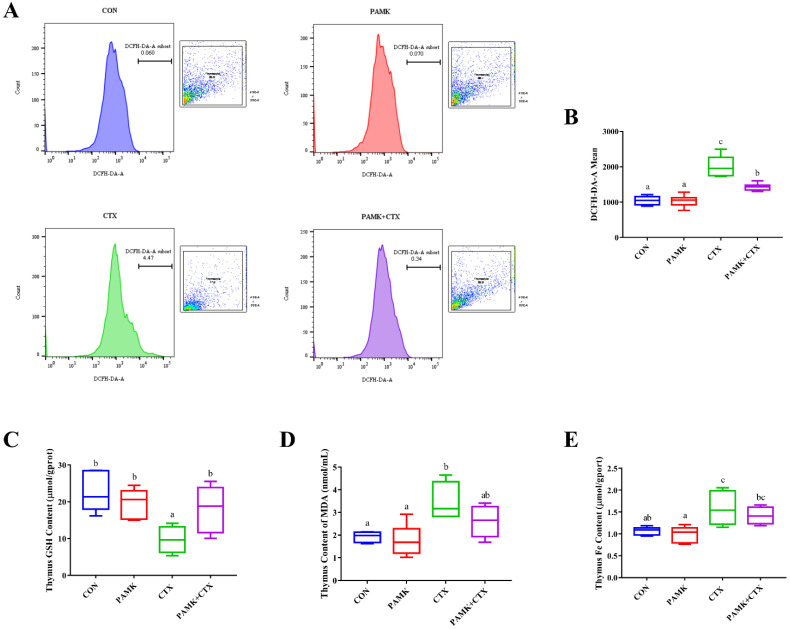
Effects of PAMK on levels of oxidative stress and iron content in thymus treated with CTX. (**A**) DCFH-DA staining to detect lipid ROS in the thymus. Mean fluorescence intensity was measured by flow cytometry; (**B**) ROS level; (**C**) GSH content; (**D**) MDA content; (**E**) iron content. Data are expressed as min to max, *n* = 10. Different letters indicate *p* < 0.05, significantly different.

**Figure 6 animals-12-03394-f006:**
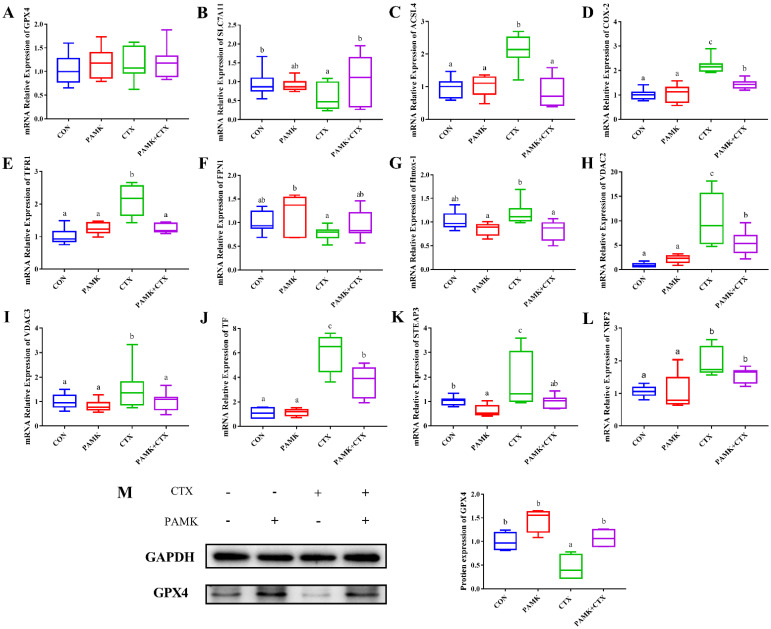
Effects of PAMK on the expression of thymus ferroptosis pathway gene treated with CTX. Relative mRNA expression of (**A**) GPX4; (**B**) SLC7A11; (**C**) ACSL4; (**D**) COX-2; (**E**) TFR1; (**F**) FPN1; (**G**) Hmox-1; (**H**) VDAC2; (**I**) VDAC3; (**J**) TF; (**K**) STEAP3;(**L**) NRF2. Protein expression of (**M**) GPX4. Data are expressed as min to max, *n* = 10. Different letters indicate *p* < 0.05, significantly different.

**Figure 7 animals-12-03394-f007:**
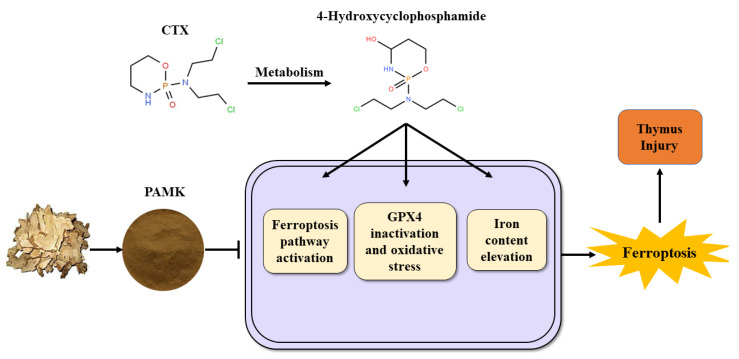
Propose mechanism of PAMK alleviation of CTX-induced thymocyte Ferroptosis in goslings.

**Table 1 animals-12-03394-t001:** Primer sequences for qPCR.

Gene	Primer (5′ → 3′)	Gen Bank Access
β-actin-F	GCACCCAGCACGATGAAAAT	XM_013174886.1
β-actin-R	GACAATGGAGGGTCCGGATT	
GPX4-F	TCGATGTGAATGGGGACGAC	XM_013200057.1
GPX4-F	GTCCTTCTCGATGACGTAGGG	
ACSL4-F	GCGGCTGAAACCCTCTTCTT	XM_013185083.1
ACSL4-R	GCCAACAGTGGACACAAGCTA	
TFR1-F	AGAATGGCTGGAGGGGTACT	XM_013195023.1
TFR1-R	TTCTCTCCAGCAGCGCATAC	
FTH1-F	ATGGTCATGGGCTTTCCCC	XM_013177583.1
FTH1-R	AATGAAGTCACACAGATGCGG	
FPN1-F	CTGGGGAGATCGTATGTGGC	XM_013178636.1
FPN1-R	AGGATGTCTGGGCCACTTTG	
Hmox-1-F	ATATGAGCACGGTCCAGCG	XM_013181078.2
Hmox-1-R	TCGTGACTATGAAGCCGAGC	
COX-2-F	TGTCCTTTCACTGCTTTCCAT	XM_013177944.1
COX-2-R	TTCCATTGCTGTGTTTGAGGT	
TF-F	ATTACTTCAGTGCGGGCTGT	XM_013186329.2
TF-R	CTCGACCAGACACCGGAAA	
VDAC2-F	GGAAGCTGCAACACGAAGAAC	XM_013176155.2
VDAC2-R	ACCAACCCAAACCCATATCCT	
VDAC3-F	CCAGTGGGGTGCTGGAATTTA	XM_013194404.2
VDAC3-R	TCCCAATGTGTTGTCCGTGT	
STEAP3-F	CCGTCAAGCAGTCCACCCT	XM_048079666.1
STEAP3-R	ACAGTACATGGGACGAGCAG	
NRF2-F	GGGATGCCCGGACATGAA	XM_013171581.2
NRF2-R	CGTCTAACTCCAGCTGAGCC	

## Data Availability

The data presented in this study are available on request from the corresponding author. The data are not publicly available due to privacy restriction.

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
