# Peer review of "Polysaccharide of Atractylodes macrocephala Koidz Alleviates Cyclophosphamide-Induced Thymus Ferroptosis in Gosling"

_animals, 2022, doi:10.3390/ani12233394_

Round 1
Reviewer 1 Report (Previous Reviewer 2)
The number of animals used in the experiment was too low to draw a reliable conclusions
Reviewer 2 Report (Previous Reviewer 1)
The aim of this study was to evaluate if the polysaccharide of Atractylodes macrocrphala Koidz (PAMK) alleviates the effects of CTX-induced thymus damage and thymocyte ferroptosis in goslings. The study is important for industry - breeders. Goose meat production is the highest in China and those researches are especially important. The biggest producer in Europe are Poland and Hungary. But this production is more than 10 times smaller than in China.
CTX induced ferroptosis and the use of PAMK as a potential antibiotic replacement is interesting. There are few article in scientific literature.
This scientific article can be a valuable supplement to the literature.My suggestion/opinion/question:
Line 26 and 92 what is BW ?
Line 24, 25 and 90 what is basic diet?
helpfull is table with abbreviation
please check the quide for authors, I think when You give first time abbreviation in text, You should give the explanaition;
in my opinion, the chapter "Conclusion" is too short. There are many results in the work that should be summarized here in order to show that PAMK can reduce the injury of thymus by alleviating the immunosuppression caused by CTX;
This manuscript is a resubmission of an earlier submission. The following is a list of the peer review reports and author responses from that submission.
Round 1
Reviewer 1 Report
I suggest creating a table of abbreviations at the beginning of the publication; it is difficult to read because many abbreviations have not been explained.
For example:
2.5.HE ->H&E what is it?
2.7.BSA? DAPI? etc.
2.8. ROS Assay? Earlier in the text there is no explanation of what ROS is
PBS
2.9. GSH, MDA, MDA is malonedialdehyde? GSH is glutatione ?
line 205 - what is SEM ? this is in statistic ? I think no, but what is it?
line 247 - Fe - this is I think iron content ? Please write "iron" not Fe, or Fe2+; Fe3+
in line 264 Author used TFR1 and in line 370 give abbreviation (please make list of abbreviation)
line 303 - TEM ? what is it?
line 360 - AA ? what is it? I think not aminoacid ? AA is generally in publications aminoacid
GPS - is glutathione peroxidase?
etc.
abstract and 2.1. what is a conventional diet? basal diet? please write 2-3 sentence about this diet
I think the abstract is unclear. After reading the text everything is clear, but reading it for the first time is very complicated. I suggested to white in abstract:
"One-day-old goslings were divided into four groups (10 goslings/group): CON, CTX, PAMK, CTX+PAMK. CON and CTX group, fed conventional diet. PAMK and CTX+PAMK groups, were fed the basal diet mixed with PAMK (400 mg/kg). Moreover for CTX and CTX+PAMK groups, given a daily injection of 40 mg/ (kg body weight) of CTX (at 19, 20, and 21 days of age). On the other hands for CON and PAMK groups, given 0.5 mL of sterilized saline into the leg muscle (at 19, 20, and 21 days of age). The goslings were fed for 28 days."
I think this is clearly.
The work is very well written. It contains valuable information. If the authors add a list of abbreviations, it will be very easy to read.
Author Response
I suggest creating a table of abbreviations at the beginning of the publication; it is difficult to read because many abbreviations have not been explained.
Response: Thanks for the reviewer’s suggestion. Done as requested. The table of abbreviations has been added as supplementary material.
For example:
2.5. HE->H&E what is it?
Response: HE is hematoxylin-eosin.
2.7. BSA? DAPI? etc.
Response: BSA is bovine serum albumin. DAPI, 4',6-diamidino-2-phenylindole, is a fluorescent dye that binds strongly to DNA and is commonly used for fluorescence microscopy observations.
2.8. ROS Assay? Earlier in the text there is no explanation of what ROS is. PBS
Response: ROS is reactive oxygen species. PBS is phosphate buffer saline.
2.9. GSH, MDA, MDA is malonedialdehyde? GSH is glutatione?
Response: Yes, they are.
line 205 - what is SEM? this is in statistic? I think no, but what is it?
Response: SEM is scanning electronic microscope.
line 247 - Fe - this is I think iron content? Please write "iron" not Fe, or Fe2+; Fe3+.
Response: Done as requested.
in line 264 Author used TFR1 and in line 370 give abbreviation (please make list of abbreviation)
Response: Done as requested.
line 303 - TEM? what is it?
Response: TEM is transmission electron microscope.
line 360 - AA? what is it? I think not aminoacid? AA is generally in publications
aminoacid
Response: AA means Arachidonic acid.
GPS - is glutathione peroxidase? etc.
Response: Yes, it is.
abstract and 2.1. what is a conventional diet? basal diet? please write 2-3 sentence about this diet
Response: This was a mistake in our writing and we have changed it all to a basic diet. And we have added a description of the basic diet as required in 2.1.
I think the abstract is unclear. After reading the text everything is clear, but reading it for the first time is very complicated. I suggested to white in abstract:
"One-day-old goslings were divided into four groups (10 goslings/group): CON, CTX, PAMK, CTX+PAMK. CON and CTX group, fed conventional diet. PAMK and CTX+PAMK groups, were fed the basal diet mixed with PAMK (400 mg/kg). Moreover, for CTX and CTX+PAMK groups, given a daily injection of 40 mg/ (kg body weight) of CTX (at 19, 20, and 21 days of age). On the other hands for CON and PAMK groups, given 0.5 mL of sterilized saline into the leg muscle (at 19, 20, and 21 days of age). The goslings were fed for 28 days."
I think this is clearly.
Response: Thanks for the reviewer’s suggestion. Done as requested.
The work is very well written. It contains valuable information. If the authors add a list of abbreviations, it will be very easy to read.

Reviewer 2 Report
General comments:
The aim of this study was to test if the polysaccharide of Atractylodes macrocrphala Koidz (PAMK) alleviates the effects of CTX-induced thymus damage and thymocyte ferroptosis in goslings. The results of the study might be interesting, especially for the breeders, as goslings, due to various stresses are susceptible to the damage of the thymus gland, resulting in immunosuppression and increased morbidity and mortality. The topic of CTX-induced ferroptosis and the use of PAMK as a potential antibiotic replacement is quite novel. The hypothesis was tested with the use of RNA-seq, Real-Time PCR, Western Blot analysis, histology, microscopic observation of morphology, immunofluorescence staining of thymus samples, and ROS, GSH, MDA, and FE assays. The authors aimed to demonstrate that PAMK can reduce the injury of thymus by alleviating the immunosuppression caused by CTX. However, in my opinion, the authors did not prove well enough that PAMK actually has such properties. First of all, the Western Blot results indicate no significant improvement in GPX protein expression after PAMK administration, we cannot even talk about a trend because, according to the data provided, p value> 0.1. This suggest that experiment should be performed on a larger number of animals. However it is unclear what number of animals in each analysis was used, n=10?. Number of animals for RNA-seq was also very little (3 per group if I understand correctly). More specific comments:
Abstract:
The description of groups is unclear, please make it more easy to understand
Please provide p values for each analisis in the abstract
Introduction:
line 56 please rewrite the sentence
line 72 please rewrite the sentence
Material and methods:
Why RNA for transcriptomic analysis was isolated from tracheal samples, while for QPCR and WB from thynus samples?
Were all analyses made from the same animal, Was there enough sample for so many analyses
What was the number of animals for each analysis?
Please provide n= in each legend of figure
Some suggestions:
Overall, good introduction, although it could be better explained why goslings are a suitable model for those studies. Is the problem with a decrease in immunity a common problem in birds living in the aquatic environment, or is it only in geese? - add the background. You used half male, half female groups, but nowhere later is gender mentioned as one of the factors. Have the results been checked for sex differences?
No software for seq analysis, or GO analysis is given.
Why was the RNA-seq of PAMK and control groups not performed?
Real-time methodology – what reagents for reaction were used?
There is some inconsistency in the narrative - the first person form, then the infinitive, etc. For
Instance as in the ‘Immunofluorescence staining’ chapter.
Figure 2, D – DEGs from which group are taken into consideration in this pathway visualization? The same figure, sections C and D are a bit illegible, it would be better to increase the format or font size of these figures.
Below is a list of examples of some minor corrections:
Line 105: roles not role
Line 57: centrifuged not centrifuge
Line 196: missing ‘the’ before ferroptosis pathway
Line 278: error in word ‘protein’
Author Response
The aim of this study was to test if the polysaccharide of Atractylodes macrocrphala Koidz (PAMK) alleviates the effects of CTX-induced thymus damage and thymocyte ferroptosis in goslings. The results of the study might be interesting, especially for the breeders, as goslings, due to various stresses are susceptible to the damage of the thymus gland, resulting in immunosuppression and increased morbidity and mortality. The topic of CTX-induced ferroptosis and the use of PAMK as a potential antibiotic replacement is quite novel. The hypothesis was tested with the use of RNA-seq, Real-Time PCR, Western Blot analysis, histology, microscopic observation of morphology, immunofluorescence staining of thymus samples, and ROS, GSH, MDA, and FE assays. The authors aimed to demonstrate that PAMK can reduce the injury of thymus by alleviating the immunosuppression caused by CTX. However, in my opinion, the authors did not prove well enough that PAMK actually has such properties. First of all, the Western Blot results indicate no significant improvement in GPX protein expression after PAMK administration, we cannot even talk about a trend because, according to the data provided, p value> 0.1. This suggest that experiment should be performed on a larger number of animals. However it is unclear what number of animals in each analysis was used, n=10?. Number of animals for RNA-seq was also very little (3 per group if I understand correctly).
Response: Thanks for the reviewer’s suggestion. For the Western Blot analysis, we analyzed six samples per group, i.e., there were a total of six bands, but the results of two of these bands were abnormal, so we felt it was reasonable to exclude the results of these two bands. When the results of these two abnormal bands were excluded, the protein expression of GPX4 was significantly increased in the PAMK+CTX group compared with the CTX group (P < 0.05). In addition, the number of animals used for the entire study was 10, but for the number of animals used for each analysis, the number of animals used varied from analysis to analysis, as answered in detail in question 6. The number of animals for RNA-seq in the original manuscript was 3, but we studied a total of 4 individuals and only selected 3 of them for analysis. Based on the reviewer's suggestion, we decided to analyze 4 individuals.
More specific comments:
Abstract:
- The description of groups is unclear, please make it more easy to understand.
Response: Thanks for the reviewer’s suggestion. Down as requested.
- Please provide p values for each analysis in the abstract.
Response: Thanks for the reviewer’s suggestion. Down as requested.
Introduction:
3、line 56 please rewrite the sentence
Response: Thanks for the reviewer’s suggestion. Done as requested. Details are in lines 56-61 of the manuscript.
4、line 72 please rewrite the sentence.
Response: Thanks for the reviewer’s suggestion. Done as requested. Details are in lines 75-77 of the manuscript.
Material and methods:
5、Why RNA for transcriptomic analysis was isolated from tracheal samples, while for QPCR and WB from thymus samples?
Response: Sorry, this was a mistake in our writing and has been changed to "thymus samples" in the manuscript.
6、Were all analyses made from the same animal, Was there enough sample for so many analyses
Response: Transcriptome sequencing: the first 4 samples of each group. Ultrastructural and immunofluorescence staining: first 3 samples of each group. Wb: 3rd-6th samples of each group. Oxidative stress levels, iron content, and RT-qPCR assays: all samples, n=10. The samples are perfectly adequate for these analyses. Because the thymus of 28-day-old goslings is in a vigorous stage of development, so each gosling has a large number of thymuses (approximately ten) and each thymus is relatively large.
7、What was the number of animals for each analysis?
Response: We used four samples per group for transcriptome sequencing and WB (WB: Among the six bands, we eliminate two bands with abnormal results, leaving four bands, i. e. n=4), three samples per group for ultrastructural observation and immunofluorescence staining, and 10 samples per group for other analyses.
8、Please provide n= in each legend of figure
Response: Down as requested.
Some suggestions:
9、Overall, good introduction, although it could be better explained why goslings are a suitable model for those studies. Is the problem with a decrease in immunity a common problem in birds living in the aquatic environment, or is it only in geese? - add the background. You used half male, half female groups, but nowhere later is gender mentioned as one of the factors. Have the results been checked for sex differences?
Response: For the poultry breeding industry, both geese and ducks suffer from the problem of reduced immunity due to living in an aquatic environment. But at present, the large-scale breeding of ducks has realized ground rearing, net breeding, cage breeding, and other completely arid breeding methods, which basically solved the problem of aquatic breeding. While the large-scale breeding of geese is currently not able to be separated from the aquatic breeding environment to achieve arid breeding, so we used goslings as model animals. ---- Relevant background has been added to the manuscript in lines 53-56 as requested by the reviewers. We did not examine the results for sex differences because this was not our focus. We used half-male and half-female goslings to control for variables so that other factors remain as consistent as possible in each group.
10、No software for seq analysis, or GO analysis is given.
Response: Seq analysis: HISAT2 and RSEM. GO analysis: Blast2GO. Thanks for the reminder. I have added it to the manuscript.
11、Why was the RNA-seq of PAMK and control groups not performed?
Response: We were interested in the differences between the CTX group and the PAMK+CTX group, specifically to understand the effect of PAMK on CTX-induced thymus injury in goslings. Therefore, RNA-seq was performed only in the CTX group and PAMK+CTX group.
12、Real-time methodology – what reagents for reaction were used?
Response: The reagent used for real-time quantification PCR assay is SYBR Green (A25742, Applied Biosystems, USA). I have added the details in lines 113-115 of the manuscript.
13、There is some inconsistency in the narrative - the first person form, then the infinitive, etc. For instance as in the ‘Immunofluorescence staining’ chapter.
Response: Revised according to the comments.
14、Figure 2, D – DEGs from which group are taken into consideration in this pathway visualization? The same figure, sections C and D are a bit illegible, it would be better to increase the format or font size of these figures.
Response: Down as requested.
15、Below is a list of examples of some minor corrections:
Line 105: roles not role
Response: Thanks to the reviewer for pointing out the error. I have revised it.
Line 57: centrifuged not centrifuge
Response: Thanks to the reviewer for pointing out the error. I have revised it.
Line 196: missing ‘the’ before ferroptosis pathway
Response: Thanks to the reviewer for pointing out the error. I have revised it.
Line 278: error in word ‘protein’
Response: Thanks to the reviewer for pointing out the error. I have revised it.
